# Meta-analysis challenges a textbook example of status signalling and demonstrates publication bias

Alfredo Sánchez-Tójar[1,2†*], Shinichi Nakagawa[3], Moisès Sánchez-Fortún[1,4‡], Dominic A Martin[2§], Sukanya Ramani[1,5], Antje Girndt[1,2], Veronika Bókony[6], Bart Kempenaers[7], András Liker[8], David F Westneat[9], Terry Burke[4], Julia Schroeder[1,2]

[1]Evolutionary Biology Group, Max Planck Institute for Ornithology, Seewiesen, Germany; [2]Department of Life Sciences, Imperial College London, Ascot, United Kingdom; [3]School of Biological, Earth and Environmental Sciences, University of New South Wales, Sidney, Australia; [4]Department of Animal and Plant Sciences, University of Sheffield, Sheffield, United Kingdom; [5]Department of Animal Behaviour, Bielefeld University, Bielefeld, Germany; [6]Lendület Evolutionary Ecology Research Group, Plant Protection Institute, Centre for Agricultural Research, Hungarian Academy of Sciences, Budapest, Hungary; [7]Department of Behavioural Ecology and Evolutionary Genetics, Max Planck Institute for Ornithology, Seewiesen, Germany; [8]MTA-PE Evolutionary Ecology Research Group, University of Pannonia, Veszprém, Hungary; [9]Department of Biology, University of Kentucky, Lexington, United States

*For correspondence:
alfredo.tojar@gmail.com

Present address: †Department of Evolutionary Biology, Bielefeld University, Bielefeld, Germany; ‡Department de Biologia Evolutiva, Ecologia i Ciències Ambientals, University of Barcelona, Barcelona, Spain; §Biodiversity, Macroecology and Biogeography, University of Goettingen, Goettingen, Germany

Competing interests: The authors declare that no competing interests exist.

**Abstract** The status signalling hypothesis aims to explain within-species variation in ornamentation by suggesting that some ornaments signal dominance status. Here, we use multilevel meta-analytic models to challenge the textbook example of this hypothesis, the black bib of male house sparrows (*Passer domesticus*). We conducted a systematic review, and obtained primary data from published and unpublished studies to test whether dominance rank is positively associated with bib size across studies. Contrary to previous studies, the overall effect size (i.e. meta-analytic mean) was small and uncertain. Furthermore, we found several biases in the literature that further question the support available for the status signalling hypothesis. We discuss several explanations including pleiotropic, population- and context-dependent effects. Our findings call for reconsidering this established textbook example in evolutionary and behavioural ecology, and should stimulate renewed interest in understanding within-species variation in ornamental traits.
DOI: https://doi.org/10.7554/eLife.37385.001

## Introduction

Plumage ornamentation is a striking example of colour and pattern diversity in the animal kingdom and has attracted considerable research (*Hill, 2002*). Most studies have focused on sexual selection as the key mechanism to explain this diversity in ornamentation (*Andersson, 1994*; *Dale et al., 2015*). The status signalling hypothesis explains within-species variation in ornaments by suggesting that these ornaments signal individual dominance status or fighting ability (*Rohwer, 1975*). Aggressive contests are costly in terms of energy use, and risk of injuries and predation (*Jakobsson et al., 1995*; *Kelly and Godin, 2001*; *Neat et al., 1998*; *Prenter et al., 2006*; *Sneddon et al., 1998*). These costs could be reduced if individuals can predict the outcome of such contests beforehand

**eLife digest** Many bird species have colourful, intricately patterned plumage. This ornamentation is generally believed to exist to attract partners. In the 1970s, however, scientists proposed an alternative idea, called the 'status signalling hypothesis'. This suggests that some birds have plumage ornaments that indicate the fighting abilities or dominance status of their bearers, much like the military badges worn by humans. These badges of status might evolve because fights, which commonly determine who gets valuable resources such as food, are a risky business. Individuals would greatly benefit from being able to predict the fighting abilities of any potential competitor and so avoid fights that they will probably lose.

Male house sparrows have a black patch on their throat, known as the bib, that has been considered to be a textbook demonstration of the status signalling hypothesis. However, most of the studies that support this idea studied small numbers of birds and used inconsistent methods. Furthermore, some recent studies have failed to replicate previous findings.

Sánchez-Tójar et al. collected data from several house sparrow populations across the world and systematically scrutinized the published literature to find all of the studies that tested the status signalling hypothesis in house sparrows. This revealed only weak evidence that the bib of male house sparrows signals the fighting abilities of its bearer. Instead, the published literature is a biased subsample; failures to replicate the hypothesis likely remain unpublished.

Currently, failures to replicate previous findings are generally deemed uninteresting, and so are not often published. By demonstrating the need to replicate findings robustly to avoid biasing conclusions, Sánchez-Tójar et al. thus join the call for a change in incentives and scientific culture.
DOI: https://doi.org/10.7554/eLife.37385.002

using so-called 'badges of status' – that is, two potential competitors could decide whether to avoid or engage in aggressive interactions based on the message provided by their opponent's signals (*Rohwer, 1975*).

Patches of ornamentation have been suggested to function as badges of status in a wide range of taxa, including insects (*Tibbetts and Dale, 2004*), reptiles (*Whiting et al., 2003*) and birds (*Senar, 2006*). The status signalling hypothesis was originally proposed to explain variation in the size of mountain sheep horns (*Beninde, 1937*; *Geist, 1966*), but the hypothesis has become increasingly important in the study of variability in plumage ornamentation in birds (*Rohwer, 1975*; *Senar, 2006*). Among the many bird species studied (*Santos et al., 2011*), the house sparrow (*Passer domesticus*) has become the classic textbook example of status signalling (*Andersson, 1994*; *Searcy and Nowicki, 2005*; *Senar, 2006*; *Davies et al., 2012*). The house sparrow is a sexually dimorphic passerine, in which the main difference between the sexes is a prominent black patch on the male's throat and chest (hereafter '*bib*'). Many studies have suggested that bib size serves as a badge of status, but most studies are based on limited sample sizes, and have used inconsistent methodologies for measuring bib and dominance status (*Nakagawa and Cuthill, 2007*; *Santos et al., 2011*).

Meta-analysis is a powerful tool to quantitatively test the overall (across-study) effect size (i.e. the '*meta-analytic mean*') for a specific hypothesis. Meta-analyses are therefore able to provide more robust conclusions than single studies and are increasingly used in evolutionary ecology (*Gurevitch et al., 2018*; *Nakagawa and Poulin, 2012a*; *Nakagawa and Santos, 2012b*; *Senior et al., 2016*). Traditional meta-analyses combine summary data across different studies, where design and methodology are study-specific (e.g. effect sizes among studies are typically adjusted for different fixed effects). These differences among studies are expected to increase heterogeneity, and therefore, the uncertainty of the meta-analytic mean (*Mengersen et al., 2013*). Meta-analysis of primary or raw data is a specific type of meta-analysis where studies are analysed in a consistent manner (*Mengersen et al., 2013*). This type of meta-analysis allows methodology to be standardized so that comparable effect sizes can be obtained across studies and is, therefore, considered the gold standard in disciplines such as medicine (*Simmonds et al., 2005*). Unfortunately, meta-analysis of primary data is still rarely used in evolutionary ecology (but see *Barrowman et al., 2003*; *Richards and Bass, 2005*; *Krasnov et al., 2009*), perhaps due to the difficulty of obtaining

the primary data of previously published studies until recently (*Culina et al., 2018*; *Schmid et al., 2003*).

An important feature of any meta-analysis is to identify the existence of bias in the literature (*Nakagawa and Santos, 2012b*; *Jennions et al., 2013*). For example, publication bias occurs whenever particular effect sizes (e.g. larger ones) are more likely found in the literature than others (e.g. smaller ones). This tends to be the case when statistical significance and/or direction of effect sizes determines whether results were submitted or accepted for publication (*Jennions et al., 2013*). Thus, publication bias can strongly affect the estimation of the meta-analytic mean, and distort the interpretation of the hypothesis (*Rothstein et al., 2005*). Several methods have been developed to identify this and other biases (*Nakagawa and Santos, 2012b*; *Jennions et al., 2013*); however, such methods are imperfect and dependent on the number of effect sizes available, and therefore should be considered as types of sensitivity analysis (*Nakagawa et al., 2017*; *Nakagawa and Santos, 2012b*).

Here, we meta-analytically assessed the textbook example of the status signalling hypothesis in the house sparrow. Specifically, we combined summary and primary data from published and unpublished studies to test the prediction that dominance rank is positively associated with bib size across studies. We found that the meta-analytic mean was small, uncertain and overlapped zero. Hence, our results challenge the status signalling function of the male house sparrow's bib. Also, we identified several biases in the published literature. Finally, we discuss potential biological explanations for our results, and provide advice for future studies testing the status signalling hypothesis.

## Results

Overall, we obtained the primary data for seven of 13 (54%) published studies, and we provided data for six additional unpublished studies (*Table 1—Appendix 1*).

### Dominance hierarchies

Mean sampling effort was 36 interactions/individual (SD = 24), which highlights that, overall, dominance hierarchies were inferred reliably across groups (*Sánchez-Tójar et al., 2018b*). The mean Elo-rating repeatability was 0.92 (SD = 0.07) and the mean triangle transitivity was 0.63 (SD = 0.28). Thus, the dominance hierarchies observed across groups of house sparrows were medium in both steepness and transitivity.

### Meta-analytic mean

Our meta-analyses revealed a small overall effect size with large 95% credible intervals that overlapped zero (*Table 2*; *Figure 1*). Additionally, the overall heterogeneity ($I^2_{overall}$) was moderate (53%; *Table 2*). Thus, our results suggested that generally, bib size is at best a weak and unreliable signal of dominance status in male house sparrows.

### Moderators of the relationship between dominance rank and bib size

None of the three biological moderators studied (season, group composition and type of interactions) explained differences among studies (*Table 3*). Sampling effort (i.e. the ratio of interactions to individuals recorded) also was not an important moderator (*Table 3*).

### Detection of publication bias

There was no clear asymmetry in the funnel plots (*Figure 2*). Also, Egger's regression tests did not show evidence of funnel plot asymmetry in any of the meta-analyses (*Table 2*). However, published effect sizes were larger than unpublished ones, and the latter were not different from zero (*Table 4*; *Figure 3*). Additionally, we found that the overall effect size decreased over time and approached zero (*Table 4*; *Figure 4*).

## Discussion

The male house sparrow's bib is not the strong across-study predictor of dominance status once believed. In contrast to the medium-to-large effect found in the previous meta-analysis (*Nakagawa et al., 2007*), our updated meta-analytic mean was small, uncertain and overlapped

**Table 1.** Studies used in the meta-analyses and meta-regressions testing the across-study relationship between dominance rank and bib size in male house sparrows.
More information is available in the data files provided (*Sánchez-Tójar et al., 2018a*).

| Study ID | Reference | Population ID | Primary data? | Number of groups* | Total number of males[†] | Comments |
|---|---|---|---|---|---|---|
| 1 | *Ritchison, 1985* | Kentucky (captivity) | No | 3 | 35 | |
| 2 | *Møller, 1987* | Denmark (wild) | Yes | 3 | 37 | |
| 3 | *Andersson and Åhlund, 1991* | Sweden (captivity) | No | 10 | 20 | Estimate originally reported as statistically non-significant. |
| 4 | *Solberg and Ringsby, 1997* | Norway (captivity) | Yes | 5 | 44 | |
| 5 | *Liker and Barta, 2001* | Hungary (captivity) | Yes | 1 | 10 | |
| 6 | *Gonzalez et al., 2002* | Spain (captivity) | No | 8 | 41 | |
| 7 | *Hein et al., 2003* | Kentucky (wild) | Yes | 4 | 39 | |
| 8 | *Riters et al., 2004* | Wisconsin (captivity) | No | 4 | 20 | |
| 9 | *Lindström et al., 2005* | New Jersey (captivity) | No | 4 | 28 | Author shared processed data, but group ID was unavailable, so data were not re-analysed. |
| 10 | *Bókony et al., 2006* | Hungary (captivity) | Yes | 2 | 19 | |
| 11 | *Buchanan et al., 2010* | Scotland (captivity) | No | 14 \n 5 | 56 \n 20 | Groups were tested twice. Post-breeding estimates originally reported as statistically non-significant. |
| 12 | *Dolnik and Hoi, 2010* | Austria (captivity) | No | 4 \n 4 | 31 \n 31 | Groups were tested twice. Pre-infection estimates originally reported as statistically non-significant. |
| 13 | *Rojas Mora et al., 2016* | Switzerland (captivity) | Yes | 14 | 56 | |
| 14 | Lendvai et al. | Hungary (captivity) | Yes[3] | 4 | 46 | Unpublished data part of: *Lendvai et al., 2004*; *Bókony et al., 2012* |
| 15 | Tóth et al. | Hungary (captivity) | Yes[3] | 3 | 35 | Unpublished data part of: *Tóth et al., 2009*; *Bókony et al., 2012* |
| 16 | Bókony et al. | Hungary (captivity) | Yes[3] | 4 | 26 | Unpublished data part of: *Bókony et al., 2010*; *Bókony et al., 2012* |
| 17 | Sánchez-Tójar et al. | Germany (captivity) | Yes[3] | 4 | 95 | Unpublished study conducted in 2014. |
| 18 | Sánchez-Tójar et al. | Lundy Island (wild) | Yes[3] | 7 | 172 | Unpublished study conducted from 2013 to 2016. |
| 19 | Westneat | Kentucky (captivity) | Yes[3] | 10 | 40 | Unpublished study conducted in 2005. |

*for primary data = yes, groups of birds containing less than four individuals were not included (see Materials and methods).

[†]Note: since most studies analysed more than one group of birds, the total number of males is different from group size in most cases (see below).

[‡]Information for the unpublished datasets is available in **Appendix 1—table 5**.

DOI: https://doi.org/10.7554/eLife.37385.003

zero. Thus, the male house sparrow's bib should not be unambiguously considered or called a badge of status. Furthermore, we found evidence for the existence of bias in the published literature that further undermines the validity of the available support for the status signalling hypothesis. First, the meta-analytic mean of unpublished studies was essentially zero, compared to the medium effect size detected in published studies. Second, we found that the effect size estimated in published studies has been decreasing over time, and recently published effects were on average no longer distinguishable from zero. Our findings call for reconsidering this textbook example in evolutionary

**Table 2.** Results of the multilevel meta-analyses on the relationship between dominance rank and bib size in male house sparrows. Additionally, the results of the Egger's regression tests are shown. Estimates are presented as standardized effect sizes using Fisher's transformation (Zr). Both meta 1 and meta 2 include published and unpublished estimates, with meta 2 including two non-reported estimates assumed to be zero (see section 'Meta-analyses').

| Meta-analysis | K | Meta-analytic mean [95% CrI] | $I^2_{population\ ID}$ [95% CrI] (%) | $I^2_{study\ ID}$ [95% CrI] (%) | $I^2_{overall}$ [95% CrI] (%) | Egger's regression [95% CrI] |
|---|---|---|---|---|---|---|
| meta 1 | 85 | 0.23 [−0.01,0.45] | 16 [0,48] | 21 [0,51] | 53 [33,73] | −0.13 [−0.59,0.27] |
| meta 2 | 87 | 0.20 [−0.01,0.40] | 15 [0,46] | 20 [0,49] | 53 [34,74] | −0.12 [−0.55,0.28] |

k = number of estimates; CrI = credible intervals; $I^2$ = heterogeneity.

DOI: https://doi.org/10.7554/eLife.37385.005

and behavioural ecology, and should stimulate renewed attention to hypotheses explaining within-species variation in ornamentation.

The status signalling hypothesis (*Rohwer, 1975*) has been extensively tested to try and explain within-species trait variation (e.g. reptiles: *Whiting et al., 2003*; insects: *Tibbetts and Dale, 2004*; humans: *Dixson and Vasey, 2012*), particularly plumage variation (*Santos et al., 2011*). Soon after the first empirical tests on birds, the black bib of male house sparrows became a textbook example of the status signalling hypothesis (*Andersson, 1994*; *Searcy and Nowicki, 2005*; *Senar, 2006*;

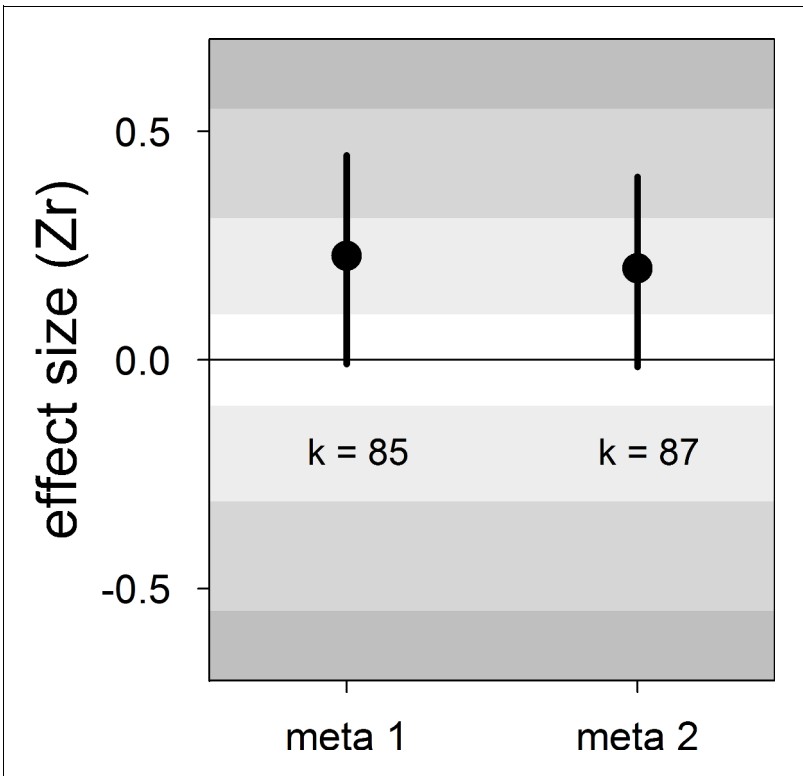

**Figure 1.** Forest plot showing the across-study effect size for the relationship between dominance rank and bib size in male house sparrows. Both meta 1 and meta 2 include published and unpublished estimates, with meta 2 including two non-reported estimates assumed to be zero (see section 'Meta-analyses'). We show posterior means and 95% credible intervals from multilevel meta-analyses. Estimates are presented as standardized effect sizes using Fisher's transformation (Zr). Light, medium and dark grey show small, medium and large effect sizes, respectively (*Cohen, 1988*). k is the number of estimates.

DOI: https://doi.org/10.7554/eLife.37385.004

**Table 3.** Results of the multilevel meta-regressions testing the effect of several moderators on the relationship between dominance rank and bib size in male house sparrows.
Estimates are presented as standardized effect sizes using Fisher's transformation (*Zr*).

| Meta-regression | Estimates | Mean [95% CrI] |
|---|---|---|
| meta 1 | intercept | 0.17 [-0.11,0.46] |
| (*k* = 85) | season | −0.11 [-0.41,0.21] |
| | group composition | 0.14 [-0.34,0.59] |
| | type of interactions | 0.33 [-0.17,0.91] |
| | $R^2_{marginal}=$ | 23 [2,48] |
| meta 2 | intercept | 0.15 [-0.10,0.45] |
| (*k* = 87) | season | −0.08 [-0.42,0.22] |
| | group composition | 0.12 [-0.32,0.62] |
| | type of interactions | 0.27 [-0.17,0.85] |
| | $R^2_{marginal}=$ | 20 [0,45] |
| sampling effort | intercept | 0.24 [-0.15,0.55] |
| (*k* = 61) | sampling effort | 0.11 [-0.49,0.74] |
| | sampling effort$^2$ | −0.14 [-0.77,0.43] |
| | $R^2_{marginal}=$ | 8 [0,24] |

*k* = number of estimates; CrI = credible intervals; $R^2_{marginal}$ = percentage of variance explained by the moderators. The factors season (non-breeding: 0, breeding: 1), group composition (mixed-sex: 0, male-only: 1), and type of interactions (all: 0, aggressive-only: 1) were mean-centred, and the covariates 'sampling effort' and its squared term were *z*-transformed.

DOI: https://doi.org/10.7554/eLife.37385.006

*Davies et al., 2012*), an idea that was later confirmed meta-analytically (*Nakagawa et al., 2007*). However, *Nakagawa et al., 2007* meta-analytic mean was over-estimated because only nine low-powered studies were available (more in *Button et al., 2013*). Here, we updated that meta-analysis with newly published and unpublished data. Our results showed that the overall effect size is much smaller and much more uncertain than previously thought. The status signalling hypothesis is thus no longer a compelling explanation for the evolution of bib size across populations of house sparrows.

Similar contradicting conclusions have been reported for other model species. An exhaustive review and meta-analysis on plumage coloration of blue tits (*Cyanistes caeruleus*) revealed that, after dozens of publications studying the function of plumage ornamentation in this species, the only robust conclusion is that females' plumage differs from that of males (*Parker, 2013*). Another example is the long-believed effect of leg bands of particular colours on the perceived attractiveness of male zebra finches (*Taeniopygia guttata*), which has been also experimentally and meta-analytically refuted (*Seguin and Forstmeier, 2012*; *Wang et al., 2018*). Finally, the existence of a badge of status in a non-bird model species, the paper wasp (*Polistes dominulus*; *Tibbetts and Dale, 2004*) has also been challenged multiple times (e.g. *Cervo et al., 2008*; *Green and Field, 2011*; *Green et al., 2013*), generating doubts about its generality. Our findings corroborate studies showing that abundant replication is needed before any strong or general conclusion can be drawn (*Aarts et al., 2015*), and highlight the existence of important impediments (e.g. publication bias) to scientific progress in evolutionary ecology (*Forstmeier et al., 2017*; *Fraser et al., 2018*).

Indeed, our results showed that the published literature on status signalling in house sparrows is likely a biased subsample. The main evidence for this is that the mean effect size of unpublished studies was essentially zero and clearly different from the mean effect size based on published studies, which was of medium size. Furthermore, this moderator (i.e. unpublished vs. published) explained a large percentage of the model's variance. In some of our own unpublished datasets, the relationship between dominance rank and bib size was never formally tested (D.F. Westneat and V. Bókony, *personal communication*, February, 2018), that is, our unpublished datasets are not all examples of the 'file drawer problem' (*sensu Rosenthal, 1979*). Egger's regression tests failed to

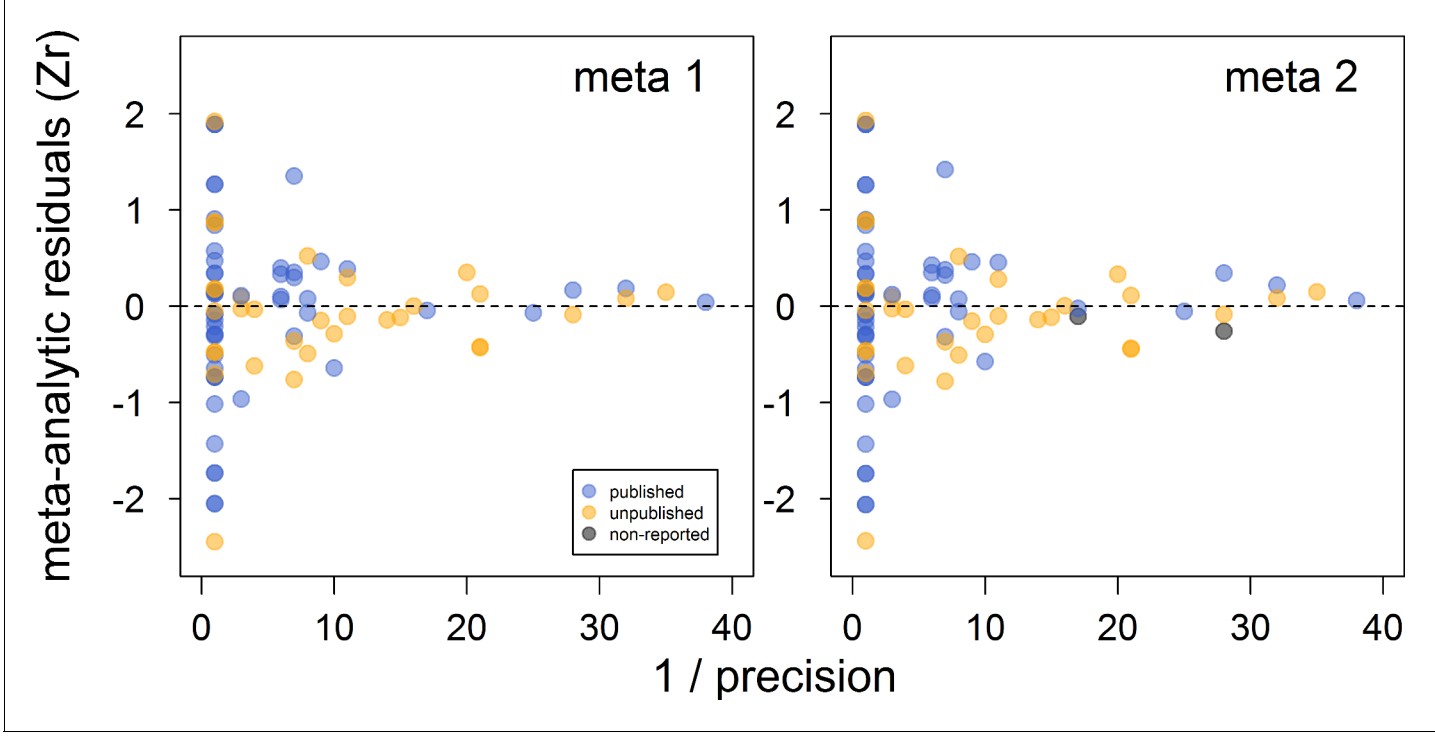

**Figure 2.** Funnel plots of the meta-analytic residuals against their precision for the meta-analyses used to test the across-study relationship between dominance rank and bib size in male house sparrows. Both meta 1 and meta 2 include published (blue) and unpublished (orange) estimates, with meta 2 including two additional non-reported estimates (grey; see section 'Meta-analyses'). Estimates are presented as standardized effect sizes using Fisher's transformation ($Zr$). Precision = square root of the inverse of the variance.

DOI: https://doi.org/10.7554/eLife.37385.007

detect any funnel plot asymmetry, even in the meta-analyses based on published effect sizes only (*Appendix 2—table 1*). However, because unpublished data indeed existed (i.e. those obtained for this study), the detection failure was likely the consequence of the limited number of effect sizes available (i.e. low power) and the moderate level of heterogeneity found in this study (*Moreno et al., 2009*; *Sterne and Egger, 2005*).

An additional type of publication bias is time-lag bias, where early studies report larger effect sizes than later studies (*Trikalinos and Ioannidis, 2005*). We detected evidence for such bias

**Table 4.** Results of the multilevel meta-regressions testing for time-lag and publication bias in the literature on status signalling in male house sparrows.

Estimates are presented as standardized effect sizes using Fisher's transformation ($Zr$). Credible intervals not overlapping zero are highlighted in bold.

| Meta-regression | Estimates | Mean [95% CrI] |
|---|---|---|
| time-lag bias | intercept | **0.26 [0.03,0.57]** |
| ($k = 53$) | year of publication | **−0.21 [-0.41,–0.01]** |
| | $R^2_{marginal}$= | 29 [0,66] |
| published *vs.* | intercept | −0.09 [-0.37,0.18] |
| unpublished ($k = 85$) | published[a] | **0.50 [0.19,0.81]** |
| | $R^2_{marginal}$= | 38 [0,68] |

$k$ = number of estimates; CrI = credible intervals; $R^2_{marginal}$ = percentage of variance explained by the moderators; [a] relative to unpublished. Year of publication was z-transformed.

DOI: https://doi.org/10.7554/eLife.37385.010

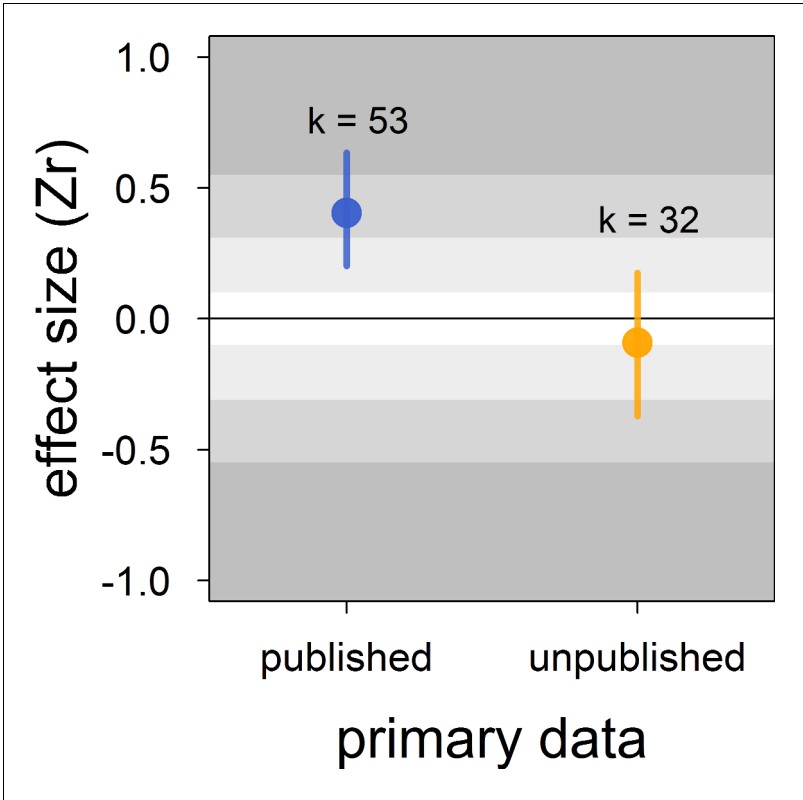

**Figure 3.** Published effect sizes for the status signalling hypothesis in male house sparrows are larger than unpublished ones. We show posterior means and 95% credible intervals from a multilevel meta-regression. Estimates are presented as standardized effect sizes using Fisher's transformation (*Zr*). Light, medium and dark grey show small, medium and large effects sizes, respectively (***Cohen, 1988***). *k* is the number of estimates.
DOI: https://doi.org/10.7554/eLife.37385.008

because the correlation between dominance rank and bib size in published studies has decreased over time and approached zero. Year of publication explained a large percentage of the model's variance, and accounting for year of publication resulted in a strong reduction of the mean effect size across published studies (***Table 4*** vs. ***Appendix 2—table 1***). Time-lag bias has been detected in other ecological studies (***Poulin, 2000***); ***Jennions and Moller, 2002b***), including a meta-analysis on status signalling across bird species (***Santos et al., 2011***). In the latter study, a positive overall (across-species) effect size persisted regardless of the time-lag bias, and no strong evidence for other types of biases was found (***Santos et al., 2011***). However, ***Santos et al., 2011*** did not attempt to analyse unpublished data, so additional evidence is needed to determine the effect that unpublished data may have on the overall validity of the status signalling hypothesis across bird species. If effect sizes based on unpublished data for other species were of similar magnitude to those obtained for house sparrows, the validity of the status signalling hypothesis across species would need reconsideration. The existence of publication bias in ecology has long been recognized (***Cassey et al., 2004***; ***Jennions and Moller, 2002b***; ***Palmer, 2000***). Publication bias leads to false conclusions if not accounted for (***Rothstein et al., 2005***), and is, thus, a serious impediment to scientific progress.

In addition to estimating the overall effect size for a hypothesis, meta-analyses are also used to assess heterogeneity among estimates (***Higgins and Thompson, 2002***; ***Higgins et al., 2003***). Understanding the sources of heterogeneity is an important step towards the correct interpretation of a meta-analytic mean, and can be done using meta-regressions (***Nakagawa and Santos, 2012b***). Here, we found that the percentage of variance that was not attributable to sampling error (i.e. heterogeneity) was moderate. This value is below the average calculated across ecological and evolutionary meta-analyses (***Senior et al., 2016***), and indicates that we accounted for large differences

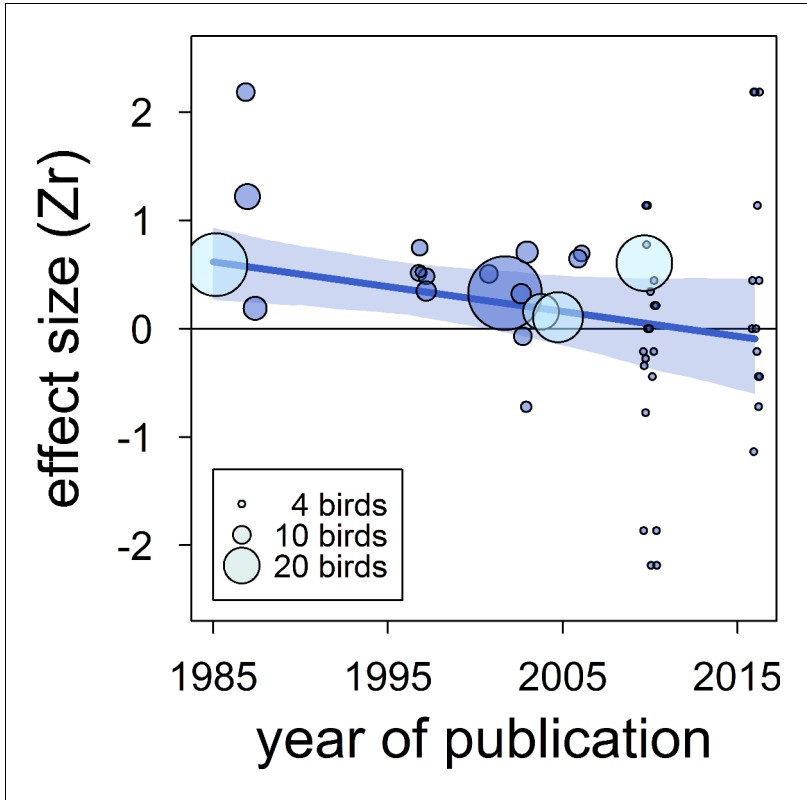

**Figure 4.** The overall published effect size for the status signalling hypothesis in male house sparrows has decreased over time since first described ($k$ = 53 estimates from 12 publications). The solid blue line represents the model estimate, and the shading shows the 95% credible intervals of a multilevel meta-regression based on published studies (see section 'Detection of publication bias'). Estimates are presented as standardized effect sizes using Fisher's transformation ($Zr$). Circle area represents the size of the group of birds tested to obtain each estimate, where light blue denotes estimates for which group size is inflated due to birds from different groups being pooled, as opposed to dark blue where group size is accurate.
DOI: https://doi.org/10.7554/eLife.37385.009

among estimates. Our meta-regressions based on biological moderators explained 20–23% of the variance (*Table 3*). However, none of the biological moderators that we tested strongly influenced the overall effect size, possibly because of limited sample sizes.

The badge of status idea is more complex than typically portrayed (reviewed by *Diep and Westneat, 2013*). Badges of status are expected to be particularly important in large and unstable groups of individuals where individual recognition would otherwise be difficult (*Rohwer, 1975*). While the evolution of badges of status in New and Old World sparrows has been related to sociality (i.e. flocking) during the non-breeding season (*Tibbetts and Safran, 2009*), additional factors need to be involved if the signal is to function in reducing aggression but retaining honesty (*Diep and Westneat, 2013*). Our results, however, did not show any evidence for a season-dependent effect as the moderator 'season' (breeding vs. non-breeding) was not a strong predictor in our models. Badges of status are expected to function both within and between sexes (*Rohwer, 1975*; *Senar, 2006*). Indeed, we found little evidence that the status signalling function of bib size differed between male-only and mixed-sex flocks. Interestingly, when competing for resources, possessing a badge of status would be beneficial for both males and females. However, male but not female house sparrows have a bib. This sexual dimorphism suggests that the bib's function is likely more important when competing for resources other than essential, *a priori* non-sex-specific resources such as food, water, sand baths and roosting sites. *Møller, 1988* and *Pape Moller, 1989* reported that female house sparrows preferentially choose males with large bibs (but see *Kimball, 1996*), and bib size has been positively correlated with sexual behaviour (*Veiga, 1996*; *Møller, 1990*), which suggests that

the bib may play a role in mate choice. Furthermore, the original status signalling hypothesis posits that the main benefit of using badges of status would be to avoid fights, which should be particularly important when interacting with unfamiliar individuals (*Rohwer, 1975*; *Senar, 2006*). Although we did not have data to test whether unfamiliarity among contestants is an important pre-requisite for the status signalling hypothesis, we found no change in mean effect size when only obviously aggressive interactions were studied. In practice, testing whether the bib is important in mediating aggression among unfamiliar individuals is difficult because the certainty of the estimates of individual dominance increases over time as more contests are recorded, but so does familiarity among contestants.

There are some additional explanations for the small and uncertain effect detected by our meta-analyses. First, different populations might be under different selective pressures regarding status signalling. Indeed, the population-specific heterogeneity ($I^2_{population\ ID}$) estimated in our meta-analyses was 15–16%, suggesting that population-dependent effects might exist. Second, although none of the moderators had a strong influence on the overall effect size, the study-specific heterogeneity estimated in our meta-analyses ($I^2_{study\ ID}$ = 20–21%) suggests that the uncertainty observed could still be explained by the status signal being context-dependent. However, context-dependence is often invoked post hoc to explain variation among studies, but strong evidence for it is lacking in most cases. Last, most studies testing the status signalling hypothesis in house sparrows are observational (*Table 1*), and the only two experimental studies conducted so far were inconclusive (*Diep, 2012*; *Gonzalez et al., 2002*). Thus, it cannot be ruled out that the weak correlation observed between dominance status and bib size is driven by a third, unknown variable. In this respect, it has been proposed that the association between melanin-based coloration (such as the bib; e.g. *Galván et al., 2015*; *Galván and Alonso-Alvarez, 2017*) and aggression is due to pleiotropic effects of the genes involved in regulating the synthesis of melanin (reviewed by *Ducrest et al., 2008*). Furthermore, bib size has been shown to correlate with testosterone, a hormone often involved in aggressive behaviour (*Gonzalez et al., 2001*) but this relationship has not been consistently observed (*Laucht et al., 2010*). Future studies should shift the focus towards understanding the function of bib size in wild populations and increase considerably the number of birds studied per group. The latter is essential because the statistical power of published tests of the status signalling hypothesis in house sparrows is alarmingly low (power = 8.5% for $r$ = 0.20, Appendix 3) and lower than the average in behavioural ecology (*Jennions, 2003*).

Our analyses have several potential limitations. First, although the number of studies included in this meta-analysis is more than double that of the previous meta-analysis (*Nakagawa et al., 2007*), it is still limited. Also, it is likely (see above) that additional unpublished data are stored in 'file drawers' (*sensu Rosenthal, 1979*). Second, most tests included in this study were still low-powered in terms of group size (median = 6 individuals/estimate, range = 4–41), and the sample size is inflated because some of the published studies pooled individuals from different groups (*Figure 4*). Third, although our results showed little evidence of an effect of sampling effort on the overall effect size, the quality of the data on dominance and bib size may still be a potential factor explaining differences among studies. Fourth, experiments will normally yield larger effect sizes than observational studies because effects of confounding factors can be reduced (*Palmer, 2000*). Nonetheless, our systematic review only identified two studies where the status signalling hypothesis was tested experimentally in house sparrows (*Gonzalez et al., 2002*; *Diep, 2012*), preventing us from estimating the meta-analytic mean for experimental studies. Note, however, that the results of those experiments were inconclusive, and potentially affected by regression to the mean (*Forstmeier et al., 2017*).

In conclusion, our results challenge an established textbook example of the status signalling hypothesis, which aims to explain within-species variation in ornament size. In house sparrows, we find no evidence that bib size consistently acts as a badge of status across studies and populations, and thus, bib size can no longer be considered a textbook example of the status signalling hypothesis. Furthermore, our analyses highlight the existence of publication biases in the literature, further undermining the validity of past conclusions. Bias against the publication of small ('non-significant') effects hinders scientific progress. We thus join the call for a change in incentives and scientific culture in ecology and evolution (*Forstmeier et al., 2017*; *Ihle et al., 2017*; *Nakagawa and Parker, 2015*; *Parker et al., 2016*).

# Materials and methods

## Systematic review

We used several approaches to maximize the identification of relevant studies. First, we included all studies reported in a previous meta-analysis that tested the relationship between dominance rank and bib size in house sparrows (*Nakagawa et al., 2007*). Second, we conducted a keyword search on Web of Science, PubMed and Scopus from 2006 to June 2017 to find studies published after *Nakagawa et al., 2007*, using the combination of keywords ['bib/badge', 'sparrow', 'dominance/status/fighting']. Third, we screened all studies on house sparrows used in a meta-analysis that tested the relationship between dominance and plumage ornamentation across species (*Santos et al., 2011*) to identify additional studies that we may have missed in our keyword search. We screened titles and abstracts of all articles and removed the irrelevant articles before examining the full texts (*Supplementary file 1*). We followed the preferred reporting items for systematic reviews and meta-analyses (PRISMA: *Moher et al., 2009*; see 'Reporting Standards Documents'). We only included articles in which dominance was directly inferred from agonistic dyadic interactions over resources such as food, water, sand baths or roosting sites (*Appendix 1—table 1*).

## Summary data extraction

Some studies had more than one effect size estimate per group of birds studied. When the presence of multiple estimates was due to the use of different statistical analyses on the same data, we chose a single estimate based on the following order of preference: (1) direct reports of effect size per group of birds studied (e.g. correlation coefficient), (2) inferential statistics (e.g. $t$, $F$ and $\chi^2$ statistics) from analyses where group ID was accounted for and no other fixed effects were included, (3) direct reports of effect size where individuals from different groups where pooled together, (4) inferential statistics from models including other fixed effects. When the presence of multiple estimates was due to the use of different methods to estimate bib size and dominance rank on the same data, we chose a single estimate per group of birds or study based on the order of preference shown in *Appendix 1—tables 1–3*. In each case, the order of preference was determined prior to conducting any statistical analysis, and thus, method selection was blind to the outcome of the analyses (more details in Appendix 1).

## Primary data acquisition

We requested primary data (i.e. agonistic dyadic interactions and bib size measures) of all relevant studies identified by our systematic review. Additionally, we asked authors to share, if available, any unpublished data that could be used to test the relationship between dominance rank and bib size in house sparrows. We emailed the corresponding author, but if no reply was received, we tried contacting all the other authors listed. One study (*Møller, 1987*) provided all primary data in the original publication and, therefore, its author was not contacted. Last, we included our own unpublished data (*Appendix 1—table 5*).

Most studies recorded data from more than one group of birds (*Table 1*). For each primary dataset obtained, we inferred the dominance hierarchy of each group of birds from the observed agonistic dyadic interactions (wins and losses) among individuals using the randomized Elo-rating method, which estimates dominance hierarchies more precisely than other methods (*Sánchez-Tójar et al., 2018b*). We then used the provided measures of individual bib size (e.g. area outlined from pictures) or, if possible, calculated bib area from length and width measures following (*Møller, 1987*). Subsequently, we estimated the Spearman's rho rank correlation ($\rho$) between individual rank and bib size for each group of birds. For one study (*Buchanan et al., 2010*), we received the already inferred dominance hierarchies for each group of birds, which we then correlated with bib size to obtain $\rho$.

## Effect size coding

Regardless of their source (primary or summary data), we transformed all estimates (e.g. $\rho$, $F$ statistics, etc) into Pearson's correlation coefficients ($r$), and then into standardized effect sizes using Fisher's transformation ($Zr$) for among-study comparison. We used the equations from *Nakagawa et al., 2007* and *Lajeunesse, 2013*. Since log(0) is undefined, $r$ values equal to 1.00 and −1.00 were transformed to 0.975 and −0.975, respectively, before calculating $Zr$. $Zr$ values of 0.100, 0.310 and 0.549

were considered small, medium and large effect sizes, respectively (equivalent benchmarks from *Cohen, 1988*). When not reported directly, the number of individuals ($n$) was estimated from the degrees of freedom. The variance in $Zr$ was calculated as: $V_{Zr} = 1/(n-3)$. Estimates ($k$) based on less than four individuals were discarded ($k$ = 33 estimates discarded).

## Meta-analyses

We ran two multilevel meta-analyses to test whether dominance rank and bib size were positively correlated across studies. The first meta-analysis, in other words '*meta 1*', included published and unpublished (re-)analysed effect sizes (i.e. effect sizes estimated from the studies we obtained primary data from), plus the remaining published effect sizes obtained from summary data (i.e. effect sizes for which primary data were unavailable).

The second meta-analysis, in other words '*meta 2*', tested the robustness of the results of meta 1 to the inclusion of non-reported estimates from studies that reported 'statistically non-significant' results without showing either the magnitude or the direction of the estimates (*Table 1*). Receipt of primary data allowed us to recover some but not all the originally non-reported estimates. Two 'non-significant' estimates were still missing. Thus, meta 2 was like meta 1 but included the two non-significant non-reported estimates, which were assumed to be zero (see *Booksmythe et al., 2017* for a similar approach). Note that non-significant estimates can be either negative or positive, and thus, assuming that they were zero may have either underestimated or overestimated them, something we cannot know from non-reported estimates. Meta-analyses based on published studies only are shown in Appendix 2.

We investigated inconsistency across studies by estimating the heterogeneity ($I^2$) from our meta-analyses following *Nakagawa and Santos, 2012b*. $I^2$ values around 25, 50% and 75% are considered as low, moderate and high levels of heterogeneity, respectively (*Higgins et al., 2003*).

## Meta-regressions

We tested if season, group composition and/or the type of interactions recorded had an effect on the meta-analytic mean. For that, we ran two multilevel meta-regressions that included the following moderators (hereafter '*biological moderators*'): (1) '*season*', referring to whether the study was conducted during the non-breeding (September-February) or the breeding season (March-August); (2) '*group composition*', referring to whether birds were kept in male-only or in mixed-sex groups; and, (3) '*type of interactions*', referring to whether the dyadic interactions recorded were only aggressive (e.g. threats and pecks), or also included interactions that were not obviously aggressive (e.g. displacements). Because only three of 19 studies were conducted in the wild ($k$ = 12 estimates; *Table 1*), we did not include a moderator testing for captive *versus* wild environments. The three biological moderators were mean-centred following *Schielzeth, 2010* to aid interpretation.

The ratio of agonistic dyadic interactions recorded to the total number of interacting individuals observed (hereafter '*sampling effort*') is a measure of sampling effort that correlates positively and logarithmically with the ability to infer the latent dominance hierarchy (*Sánchez-Tójar et al., 2018b*). The higher this ratio, the more precisely the latent hierarchy can be inferred (*Sánchez-Tójar et al., 2018b*). For the subset of studies for which the primary data of the agonistic dyadic interactions were available (12 out of 19 studies; *Table 1*), we ran a multilevel meta-regression including sampling effort and its squared term as *z*-transformed moderators (*Schielzeth, 2010*). The squared term was included because of the observed logarithmic relationship between sampling effort and the method's performance (*Sánchez-Tójar et al., 2018b*). This meta-regression tested whether sampling effort had an effect on the meta-analytic mean: (i) a positive estimate would indicate that the meta-analytic mean may have been affected by the inclusion of studies with unreliable estimates of dominance rank. In contrast, (ii) a negative estimate would indicate that effect sizes were larger when based on unreliable estimates of dominance rank and hence provide evidence for the existence of publication bias.

For all meta-regressions, we estimated the percentage of variance explained by the moderators ($R^2_{marginal}$) following (*Nakagawa and Schielzeth, 2013*).

## Random effects

All meta-analyses and meta-regressions included the two random effects '*population ID*' and '*study ID*'. Population ID was related to the geographical location of the population of birds studied. We used Google maps to estimate the distance over land (i.e. avoiding large water bodies) among populations, and assumed the same population ID when the distance was below 50 km (13 populations; *Table 1*). Study ID encompassed those estimates obtained within each specific study (19 studies). Two studies tested the prediction twice for the same groups of birds (*Table 1*) and, within each population, some individuals may have been sampled more than once. However, we could not include 'group ID' and/or 'individual ID' as additional random effects due to either limited sample size or because the relevant data were not available.

## Detection of publication bias

For the meta-analyses, we assessed publication bias using two methods that are based on the assumption that funnel plots should be symmetrical. First, we visually inspected asymmetry in funnel plots of meta-analytic residuals against the inverse of their precision (defined as the square root of the inverse of $V_{Zr}$) for each meta-analysis. Funnel plots based on meta-analytic residuals (the sum of effect-size-level effects and sampling-variance effects) are more appropriate than those based on effect sizes when multilevel models are used (*Nakagawa and Santos, 2012b*). Second, we ran Egger's regressions using the meta-analytic residuals as the response variable, and the precision (see above) as the moderator (*Nakagawa and Santos, 2012b*) for each meta-analysis. If the intercept of such a regression does not overlap zero, estimates from the opposite direction to the meta-analytic mean might be missing and hence we consider this evidence of publication bias (*Nakagawa and Santos, 2012b*). Further, we tested whether published estimates differed from unpublished estimates. For that, we ran a multilevel meta-regression that included population ID and study ID as random effects, and '*unpublished*' (two levels: yes (0), no (1)) as a moderator. This meta-regression was based on meta 1 (i.e. it did not include the two non-reported estimates). We did not use the trim-and-fill method (*Duval and Tweedie, 2000a*; *Duval and Tweedie, 2000b*) because this method has been advised against when significant heterogeneity is present (*Moreno et al., 2009*; *Jennions et al., 2013*), as it was the case in our meta-analyses (see section 'Results').

Finally, we analysed temporal trends in effect sizes that could indicate 'time-lag bias'. Time-lag bias is common in the literature (*Jennions and Moller, 2002b*; *Poulin, 2000*), and occurs when the effect sizes of a specific hypothesis are negatively correlated with publication date (i.e. effect sizes decrease over time; *Trikalinos and Ioannidis, 2005*). A decrease in effect size over time can have multiple causes. For example, initial effect sizes might be inflated due to low statistical power ('winner's curse') but published more easily and/or earlier due to positive selection of statistically significant results (reviewed by *Koricheva et al., 2013*). We ran a multilevel meta-regression based on published effect sizes only, where '*year of publication*' was included as a *z*-transformed moderator (*Nakagawa and Santos, 2012b*).

All analyses were run in R v. 3.4.0 (*R Core Team, 2017*). We inferred individual dominance ranks from agonistic dyadic interactions using the randomized Elo-rating method from the R package 'aniDom' v. 0.1.3 (*Farine and Sánchez-Tójar, 2017*; *Sánchez-Tójar et al., 2018b*). Additionally, we described the dominance hierarchies observed in the groups of house sparrows for which primary data was available. For that we estimated the uncertainty of the dominance hierarchies using the R package 'aniDom' v. 0.1.3 (*Farine and Sánchez-Tójar, 2017*; *Sánchez-Tójar et al., 2018b*) and the triangle transitivity (*McDonald and Shizuka, 2013*) using the R package 'compete' 3.1.0 (*Curley, 2016*). We used the R package 'MCMCglmm' v. 2.24 (*Hadfield, 2010*) to run the multilevel meta-analytic (meta-regression) models (*Hadfield and Nakagawa, 2010*). For each meta-analysis and meta-regression, we ran three independent MCMC chains for 2 million iterations (thinning = 1,800, burn-in = 200,000) using inverse-Gamma priors (V = 1, nu = 0.002). Model chains were checked for convergence and mixing using the Gelman-Rubin statistic. The auto-correlation within the chains was <0.1 in all cases. For each meta-analysis and meta-regression, we chose the model with the lowest DIC value to extract the posterior mean and its 95% highest posterior density intervals (hereafter 95% credible interval). We report all data exclusion criteria applied and the results of all analyses conducted in our study.

## Data and code availability

We provide all of the R code and data used for our analyses (*Sánchez-Tójar et al., 2018a*).

## Acknowledgements

AST and AG are grateful for the support of the International Max Planck Research School (IMPRS) for Organismal Biology. We thank Katherine L Buchanan, Sanh K Diep, Fabrice Helfenstein, Anna Kulcsár, Ádám Z Lendvai, Karin M Lindström, Thor Harald Ringsby, Alfonso Rojas Mora, Bernt-Erik Sæther, Emmi Schlicht, Erling J Solberg, Zoltán Tóth and Jarle Tufto for providing the primary data of published and unpublished studies. We thank Wolfgang Forstmeier, Lucy Winder, and Tim Parker and an anonymous reviewer for constructive feedback on the manuscript.

## Additional information

### Funding

| Funder | Grant reference number | Author |
| --- | --- | --- |
| Max-Planck-Gesellschaft | Open-access funding | Alfredo Sánchez-Tójar |
| Max-Planck-Gesellschaft | Funding captive house sparrow population | Bart Kempenaers |
| National Science Foundation | | David F Westneat |
| Natural Environment Research Council | NE/N013832/1 | Terry Burke |
| Volkswagen Foundation | | Julia Schroeder |
| H2020 Marie Skłodowska-Curie Actions | CIG PCIG12-GA-2012-333096 | Julia Schroeder |

The funders had no role in study design, data collection and interpretation, or the decision to submit the work for publication.

### Author contributions

Alfredo Sánchez-Tójar, Conceptualization, Data curation, Software, Formal analysis, Validation, Investigation, Visualization, Methodology, Writing—original draft, Project administration, Writing—review and editing; Shinichi Nakagawa, Conceptualization, Software, Supervision, Writing—review and editing; Moisès Sánchez-Fortún, Dominic A Martin, Sukanya Ramani, Antje Girndt, Veronika Bókony, Bart Kempenaers, András Liker, David F Westneat, Investigation, Writing—review and editing; Terry Burke, Julia Schroeder, Conceptualization, Supervision, Funding acquisition, Writing—review and editing

### Author ORCIDs

Alfredo Sánchez-Tójar (iD) http://orcid.org/0000-0002-2886-0649
Shinichi Nakagawa (iD) http://orcid.org/0000-0002-7765-5182
Dominic A Martin (iD) https://orcid.org/0000-0001-7197-2278
Antje Girndt (iD) http://orcid.org/0000-0002-9558-1201
Veronika Bókony (iD) http://orcid.org/0000-0002-2136-5346
Bart Kempenaers (iD) http://orcid.org/0000-0002-7505-5458
David F Westneat (iD) http://orcid.org/0000-0001-5163-8096
Terry Burke (iD) https://orcid.org/0000-0003-3848-1244

### Decision letter and Author response

Decision letter https://doi.org/10.7554/eLife.37385.028
Author response https://doi.org/10.7554/eLife.37385.029

# Additional files

## Supplementary files

• Supplementary file 1. Decision spreadsheet of the systematic review.
DOI: https://doi.org/10.7554/eLife.37385.011

• Transparent reporting form
DOI: https://doi.org/10.7554/eLife.37385.012

• Reporting standard 1. PRISMA statement.
DOI: https://doi.org/10.7554/eLife.37385.013

## Data availability

All data generated or analysed during this study are openly available at the Open Science Framework. We direct the reader to this project in the main text and the reference list. Link: https://osf.io/cwkxb/ DOI: 10.17605/OSF.IO/CWKXB

The following dataset was generated:

| Author(s) | Year | Dataset title | Dataset URL | Database and Identifier |
|---|---|---|---|---|
| Alfredo Sánchez-Tójar, Shinichi Nakagawa, Moisès Sánchez-Fortún, Dominic A Martin, Sukanya Ramani, Antje Girndt, Veronika Bókony, Bart Kempenaers, András Liker, David F Westneat, Terry Burke, Julia Schroeder | 2018 | Supporting information for "Meta-analysis challenges a textbook example of status signalling and demonstrates publication bias" | http://doi.org/10.17605/OSF.IO/CWKXB | Open Science Framework, 10.17605/OSF.IO/CWKXB |

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

## Appendix 1

DOI: https://doi.org/10.7554/eLife.37385.014

# Information about data used in the study

**Appendix 1—table 1. Summary of key differences in methodology among all studies (published and unpublished) testing the relationship between dominance rank and bib size in male house sparrows (N = 19 studies).**

| Variable | Levels | Number of studies | Order of preference* |
|---|---|---|---|
| Group composition | Males and females | 11 | - |
| | Males only | 8 | - |
| Resource competed for | Food only | 12 | - |
| | Food, water and roosting place | 6 | - |
| | Females | 1 | - |
| Type of interactions | Aggressive only | 12 | - |
| | Aggressive and non-aggressive | 7 | - |
| Interactions recording protocol | Live observations | 11 | - |
| | Video | 6 | - |
| | Live and video observations | 2 | - |
| Type of bib size measured | Visible | 14 | 1 |
| | Hidden | 2 | 2 |
| | Both | 3 | - |
| Beak angle during measurement | 90 deg | 8 | 1 |
| | 180° | 3 | 2 |
| | Both | 1 | - |
| | Unknown | 7 | - |
| Season | Non-breeding | 13 | - |
| | Breeding | 5 | - |
| | Both | 1 | - |
| Study location | Captive | 16 | - |
| | Wild | 2 | - |
| | Both | 1 | - |

*Order of preference used for the analyses (see main text). The order of preference was determined based on how frequently the method was used in previous studies.

DOI: https://doi.org/10.7554/eLife.37385.015

**Appendix 1—table 2.** List of the different methods used to estimate bib size in all studies (published and unpublished) testing the relationship between dominance rank and bib size in male house sparrows (N = 19 studies). Note that some studies used more than one method to estimate bib size.

| Method to estimate bib size | Number of times used | Order of preference‡ |
|---|---|---|
| Area* | 8 | 1 |
| *Møller, 1987*'s equation | 6 | 2 |
| Length and width† | 3 | 2 |

*Appendix 1—table 2 continued on next page*

*Appendix 1—table 2 continued*

| Method to estimate bib size | Number of times used | Order of preference‡ |
|---|---|---|
| Length only | 2 | 3 |
| *Møller, 1987*'s drawings | 1 | 4 |
| *Veiga, 1993*'s equation | 1 | 5 |

*Area was measured from pictures (N = 5 studies), by tracing and weighing (N = 2 studies), and by tracing and ranking (N = 1 study).

†If length and width were available, we estimated bib area using *Møller, 1987*'s equation.

‡Order of preference used for the analyses (see main text). The order of preference was determined based on how frequently the method was used in previous studies.

DOI: https://doi.org/10.7554/eLife.37385.016

**Appendix 1—table 3.** List of the different methods used to infer dominance rank from dyadic interactions in published studies that tested the relationship between dominance rank and bib size in male house sparrows (N = 13 published studies, 11 different methods). Note that some studies used more than one method to estimate dominance rank and that unpublished studies are not included in this summary.

| Method to infer dominance rank | Number of times used | Order of preference* |
|---|---|---|
| Proportion of contests won | 4 | 4 |
| Proportion of initiated contests | 3 | 5 |
| Kendall's linearity index | 2 | 3 |
| Proportion of contests won per dyad | 2 | 6 |
| Proportion of initiated contests won | 2 | 6 |
| David's score | 1 | 1 |
| I and SI | 1 | 2 |
| Landau's linearity index | 1 | 3 |
| Proportion of the received attacks won | 1 | 7 |
| Proportion of birds dominated | 1 | 7 |
| Proportion of contests won per dyad + linear assumption | 1 | 7 |

*Order of preference used for the analyses (see main text). The order of preference was determined based on both how frequently the method was used in previous studies and by taking into account the (expected) performance of each of the methods. First, higher order of preference was assigned to methods specifically designed for inferring linear dominance hierarchies (i.e. David's score, I and SI, Landau's and Kendall's linearity indices). We used the information available in *Sánchez-Tójar et al., 2018b* to rank David's score and I and SI as first and second methods in preference, respectively. Second, we ranked the remaining (proportion-based) methods based on how frequently they were used in previous studies. Importantly, the order of preference was chosen prior to conducting any statistical analysis, and thus, method selection was blind to the outcome of the analyses.

DOI: https://doi.org/10.7554/eLife.37385.017

**Appendix 1—table 4.** Additional comments on some of the published studies included in the meta-analysis.

| Reference | Comments |
|---|---|
| *Ritchison, 1985* | According to the original publication, the total number of birds studied was 35, as opposed to the 25 individuals used in the meta-analyses of *Nakagawa et al., 2007* and *Santos et al., 2011*. |

*Appendix 1—table 4 continued on next page*

*Appendix 1—table 4 continued*

| Reference | Comments |
|---|---|
| *Hein et al., 2003* | The total number of birds included in our re-analysis of the primary data is smaller than that presented in the original publication. This is because our re-analysis only included fully identified individuals (e.g. birds missing rings could not be included). |
| *Dolnik and Hoi, 2010* | 32 males were selected for the experiment, but one bird was excluded before the start of the experiment. Thus, *n* was set to 31 individuals for this study. |
| *Buchanan et al., 2010* | 96 birds were separated in 24 aviaries of four individuals each. The final *n* of several aviaries was less than four individuals, and therefore, these aviaries were not included in our meta-analyses (see main text, section 'Materials and Methods'). |
| *Rojas Mora et al., 2016* | According to the primary data, one male did not interact, and thus, *n* was set to 59 individuals in Appendix 2. |

DOI: https://doi.org/10.7554/eLife.37385.018

**Appendix 1—table 5.** Data descriptions for the unpublished data analysed in the meta-analysis.

| Study ID* | Data description |
|---|---|
| 14 | 88 individuals were separated into four captive mixed-sex groups. Live observations after mild food deprivation were conducted to record agonistic dyadic interactions (i.e. fights) over (mostly) food for around one week in Feb 2003 (total = 1,563 fights). Bib length and width were measured for each male before the dominance observations using a ruler. More information can be found in *Lendvai et al., 2004* and *Bókony et al., 2012*. |
| 15 | 61 individuals were separated into three captive mixed-sex groups. Live observations after mild food deprivation were conducted to record agonistic dyadic interactions (i.e. fights) over (mostly) food between Oct and Dec 2005 (two groups) and 2006 (one group; total = 2,003 fights). Bib area was measured for each male using standardized pictures taken after the dominance observations. More information can be found in *Tóth et al., 2009* and *Bókony et al., 2012*. |
| 16 | 60 individuals were separated into four captive mixed-sex groups. Live and video observations after mild food deprivation were conducted to record agonistic dyadic interactions (i.e. fights) over (mostly) food for around two weeks per group between Oct 2007 and Feb 2008 (total = 6,641 fights). Bib length and width were measured for each male before the dominance observations using a ruler. More information can be found in *Bókony et al., 2010* and *Bókony et al., 2012*. |
| 17 | 96 males were separated into four captive male-only groups. Videos after mild food deprivation were taken to record agonistic dyadic interactions (i.e. fights) over food for 10 days between Oct and Dec 2014 (total = 3,776 fights). Bib area was measured several times for each male (median = 3 times/male, range = 2 to 6) using standardized pictures taken from Oct to Dec 2014, and the mean bib area of each individual was used in the analyses. |
| 18 | 453 individuals (215 females and 238 males) were observed in seven discrete sampling events in a wild population of house sparrows at Lundy Island, UK. Videos were taken to record agonistic dyadic interactions (i.e. fights) over food for 20 days between Nov 2013 and Dec 2016 (total = 11,063 fights). Bib length was measured several times for each male (median = 1 time/male, range = 1 to 6) from Nov 2013 to Dec 2016 using a calliper, and the mean bib area of each individual in each sampling event was used in the analyses. |
| 19 | 128 individuals were separated into 16 captive mixed-sex groups. Live observations after mild food deprivation were conducted to record agonistic dyadic interactions (i.e. supplants and hold-offs) over food between Mar and Apr 2005 (total = 5,496 fights). Bib length and width were measured for each male before the dominance observations using a calliper as in *Morrison et al., 2008*. |

*Study ID corresponding to *Table 1* in main text.

DOI: https://doi.org/10.7554/eLife.37385.019

## Appendix 2

DOI: https://doi.org/10.7554/eLife.37385.020

### Meta-analyses based on published studies only

**Appendix 2—table 1. Results of two multilevel meta-analyses to test the relationship between dominance rank and bib size in male house sparrows based on published studies only.** Published 1 includes published effect sizes obtained from summary data, whereas published 2 includes published re-analysed effect sizes together with the remaining published effect sizes obtained from summary data. Additionally, the results of the Egger's regressions are shown. Estimates are presented as standardized effect sizes using Fisher's transformation ($Zr$). Credible intervals not overlapping zero are highlighted in bold.

| Meta-analysis | K | Meta-analytic mean [95% CrI] | $I^2_{population\ ID}$ [95% CrI] (%) | $I^2_{study\ ID}$ [95% CrI] (%) | $I^2_{overall}$ [95% CrI] (%) | Egger's regression [95% CrI] |
|---|---|---|---|---|---|---|
| Published 1 | 20 | 0.45 [0.26,0.63] | 17 [0,51] | 17 [0,53] | 46 [15,78] | 0.42 [−0.73,1.48] |
| Published 2 | 53 | 0.40 [0.11,0.67] | 14 [0,46] | 13 [0,42] | 46 [17,72] | −0.25 [−0.73,0.26] |

$k$ = number of estimates; CrI = credible intervals; $I^2$ = heterogeneity.

DOI: https://doi.org/10.7554/eLife.37385.021

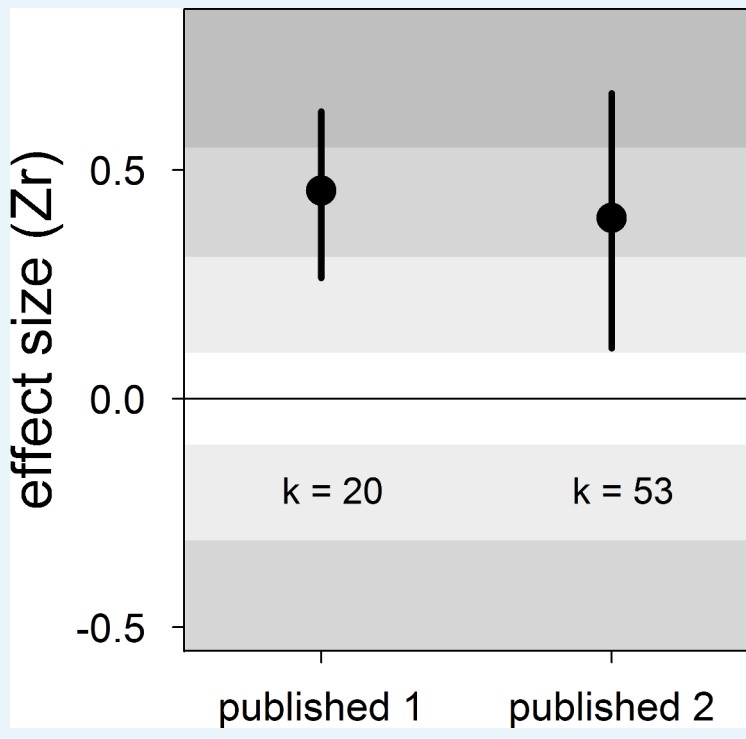

**Appendix 2—figure 1.** Forest plot showing the overall effect size of the relationship between dominance rank and bib size in male house sparrows based on published studies only. Published 1 includes published effect sizes obtained from summary data, whereas published 2 includes published re-analysed effect sizes together with the remaining published effect sizes obtained from summary data. We show posterior means and 95% credible intervals from multilevel meta-analyses. Estimates are presented as standardized effect sizes using

Fisher's transformation (*Zr*). Light, medium and dark grey show small, medium and large effect sizes, respectively (***Cohen, 1988***). *k* is the number of estimates.
DOI: https://doi.org/10.7554/eLife.37385.022

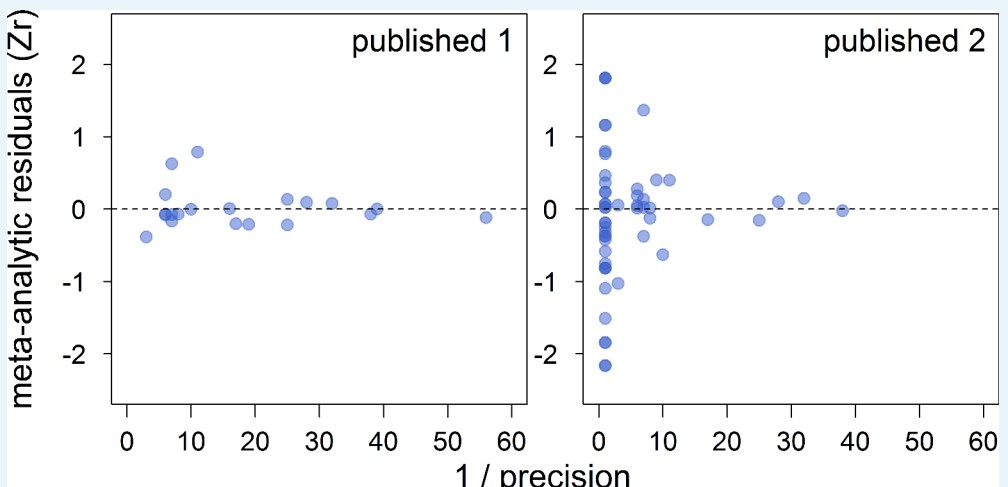

**Appendix 2—figure 2.** Funnel plots of the meta-analytic residuals against their precision for the meta-analyses based on published studies only. Published 1 includes published effect sizes obtained from summary data, whereas published 2 includes published re-analysed effect sizes together with the remaining published effect sizes obtained from summary data. Estimates are presented as standardized effect sizes using Fisher's transformation (*Zr*). Precision = square root of the inverse of the variance.
DOI: https://doi.org/10.7554/eLife.37385.023

# Appendix 3

DOI: https://doi.org/10.7554/eLife.37385.024

## Power analysis based on the estimated meta-analytic mean

### R code used and explanations:

First, we need to clear up the memory and load the pwr library.

```
# clear memory

rm(list = ls())

# package needed

library(pwr)
```

Furthermore, we created a function to transform Zr values into r values. This is because our meta-analyses were based on Zr values, but the power analysis is based on r values.

```
# function to convert Zr to r

Zr.to.r<-function(Zr){

r<-(exp(2*Zr)−1)/(exp(2*Zr)+1)

}
```

Power analysis

Next, we estimated the sample size necessary to find an effect size as small as the one estimated by our meta-analysis (Zr = 0.20). We used a significance level of 0.05, and the recommended 80% statistical power (*Cohen, 1988*).

```
pwr.r.test(r = Zr.to.r(0.20), sig.level = 0.05, power = 0.8)

##

##   approximate correlation power calculation (arctangh transformation)

##

##     n = 198.3401

##     r = 0.1973753

##   sig.level = 0.05

##     power = 0.8
```

```
##   alternative = two.sided
```

This shows that we would need the dominance rank and bib size of 198 individuals to find a significant r correlation of 0.20 with an 80% statistical power.

Additionally, we estimated the across-study statistical power of the tests on status signalling in house sparrows to compare it to the overall statistical power found in the behavioural ecology literature (*Jennions, 2003*).

```
pwr.r.test(n = 10, r = Zr.to.r(0.20), sig.level = 0.05)

##

##   approximate correlation power calculation (arctangh transformation)

##

##      n = 10

##      r = 0.1973753

##  sig.level = 0.05

##    power = 0.08474157

##   alternative = two.sided
```

This shows that the statistical power of the sparrow literature on status signaling is as low as 8.5%, which is alarming.

