## [Decision Letter]

Thank you for submitting your article "Meta-analysis challenges a textbook example of status signalling and demonstrates publication bias" for consideration by *eLife*. Your article has been reviewed by two peer reviewers, and the evaluation has been overseen by Diethard Tautz as the Senior and Reviewing Editor.

The reviewers have discussed the reviews with one another and the Reviewing Editor has drafted this decision to help you prepare a revised submission.

Summary:

This study updates a previous meta-analysis of the correlation between a plumage ornament (bib size) and dominance status in house sparrows, using meta-analysis of the primary data from a larger sample of published and unpublished studies. House sparrows have been considered an exemplar for the 'badge of status' hypothesis to explain the evolution of male ornaments, and so this is a particularly important analysis. The present analyses find a small mean effect size with 95% credible intervals overlapping zero, indicating there is no association between bib size and dominance status across studies. They also find that the mean effect size among published studies is significantly larger than among unpublished studies, and that the mean effect size declines over time, both suggesting potential publication biases.

These results are quite a striking refutation of a previously well-accepted hypothesis, and provide clear indication that a range of biases in the publication process may lend unwarranted support to many hypotheses circulating in the literature.

The study appears thorough and well implemented. Further, the authors show a commendable degree of transparency regarding their process.

However, there are a number of points that need attention and better description before the paper can be published.

Essential revisions:

1) How did the authors choose which relationships to include in their analyses? It is not adequately clear that the authors took sufficient steps to avoid bias in which effect sizes they chose to include. Thus, we recommend that they include more explanation of how they avoided this bias, or, if the risk of bias is plausible, then a re-analysis designed to avoid bias is required.

2) Please consider the following concerns about assessment of publication bias.a) To what extent might your re-analysis of raw data have concealed publication bias? Were you using your re-analyzed data for the funnel plots and Egger regression? This was not clear from the methods.

b) On a related note, using meta-analytic residuals to test for publication bias assumes that none of the modeled variables correlate with publication bias. Is that reasonable in this case?

c) If publication bias towards strong effects were at work, we would expect to observe the strongest effects from those studies with greater sampling error (those with lower sampling effort). The observed absence of this result is what we would expect with little or weak publication bias. This should be acknowledged (though see points a) and b) above). In contrast, there is insufficient explanation of why the results in Figure 4 (the change in published effects over time) are strong evidence of publication bias.

3) Please state also the effect of using meta-analysis of the raw data, reanalysed in a consistent way, compared to using calculated effect sizes or summary statistics available in the published studies. While use of raw data does seem like a desirable standard to strive for in meta-analysis, it doesn't seem to make that big of a difference when comparing the results of 'published 1' and 'published 2' in Appendix—Table 6. Please comment whether you see this as a priority in a list of recommendations for best practice in meta-analysis (or best practice in the production of primary studies that can be effectively included in meta-analyses) or do the many other potential sources of bias have stronger effects on the confidence in/conclusions that can be drawn from meta-analytic estimates?

4) It is necessary to add a statement (and explanatory text – compare https://osf.io/hadz3/) to the paper confirming whether, for all questions, you have reported all measures, conditions, data exclusions, and how you determined sample sizes.

5) Please also address the following editorial point:

The Abstract contains a statement about "the validity of the current scientific publishing culture". Similar statements are made in the main text (Introduction, last paragraph, Discussion, first and last paragraphs), but the manuscript never goes into detail about these matters.

Please, therefore, delete the following passage from the Abstract: "raise important concerns about the validity of the current scientific publishing culture". Please also delete the corresponding statements in the last paragraph of the Introduction and the first paragraph of the Discussion. It is fine to keep the statement in the last paragraph of the Discussion.

---

## [Author Response]

Essential revisions:1) How did the authors choose which relationships to include in their analyses? It is not adequately clear that the authors took sufficient steps to avoid bias in which effect sizes they chose to include. Thus, we recommend that they include more explanation of how they avoided this bias, or, if the risk of bias is plausible, then a re-analysis designed to avoid bias is required.

We thank the reviewers for spotting this lack of transparency in our writing. We have now included explanations about how those orders of preference were determined (see Appendix 1, Appendix—tables 1-3).

To re-iterate this here in short, we took the necessary steps to avoid any bias by deciding the order of preference prior to conducting any analysis. Indeed, we did not run additional analyses based on different orders of preference, and thus, our study does not suffer from selective reporting of results.

For the methods shown in Appendix 1—tables 1 and 2, the order of preference was determined by how often the methods were used in previous studies. This decision was based on our attempt to standardize among-study methodology as much as possible.

For the methods shown in Appendix 1—table 3 (i.e. methodology to infer dominance rank), the order of preference was determined by both method performance and frequency of use. We divided the methods in two groups: (i) “linearity-based” (i.e. David’s score, I&SI, Landau’s and Kendall’s indices), and (ii) “proportion-based” methods. The first group contained methods that are either based on finding the hierarchy that best approaches linearity or that consider the strength of the opponents to infer individual success. Linearity-based methods are expected to outperform the proportion-based methods, which are based on simple proportions of contests won/lost per individual. Thus, we gave priority to the linearity-based methods, and ranked them using the results from Sánchez-Tójar et al., 2018, (see Figure 5 of that paper). The methods from the second group were then ranked based on how often they were used in previous studies. Finally, in the analyses that involved primary data, the randomized Elo-rating was prioritized due to its higher performance (Sánchez-Tójar et al., 2018) and used to infer the dominance hierarchy of all studies for which primary data were available (i.e. 12 out of 19 studies).

In addition to the explanations added in Appendix 1—tables 1-3, we have also added the following statement at the end of the “Summary data extraction” subsection:

**“**In each case, the order of preference was determined prior to conducting any statistical analysis, and thus, method selection was blind to the outcome of the analyses (more details in Appendix 1).”

2) Please consider the following concerns about assessment of publication bias.a) To what extent might your re-analysis of raw data have concealed publication bias? Were you using your re-analyzed data for the funnel plots and Egger regression? This was not clear from the methods.

We thank the reviewers for spotting this lack of clarity in our writing. We indeed explored publication bias for each meta-analysis by running an Egger’s regression and generated a funnel plot for each of the four meta-analyses in our manuscript (Tables 2 and Appendix 2—table 6, Figures 2 and Appendix 2—figure 2). The (re-)analysed data were used in all but the meta-analysis called “published 1” (Appendix 2—table 6, Appendix 2—figure 1-2), which was based on the published original effect sizes only, and thus free from any potential concealment due to our (re-)analysis. The results of that meta-analysis agreed with those of the other meta-analyses, i.e. publication bias was neither apparent from visually inspecting funnel plots nor from the results of the Egger’s regressions. However, this is likely due to the difficulty of detecting publication bias when the number of effect sizes is limited and heterogeneity is present (see Moreno et al., 2009).

Overall, if (re-)analysing led to an increase in heterogeneity, detecting publication bias via funnel plots and Egger’s regressions could be more difficult. However, mean total heterogeneity (*I^2^*_overall_) did not increase when including (re-)analysed effect sizes (Appendix 2—table 6, “published 1” vs. “published 2”).

Importantly, our (re-)analysis was a necessary step to show and account for the real heterogeneity among effect sizes. We have clarified the methodology by writing “for each meta-analysis” at the end of the following sentences:

“First, we visually inspected asymmetry in funnel plots of meta-analytic residuals against the inverse of their precision (defined as the square root of the inverse of *V_Zr_*) for each meta-analysis.”

“Second, we ran Egger’s regressions using the meta-analytic residuals as the response variable, and the precision (see above) as the moderator (Nakagawa and Santos, 2012) for each meta-analysis.”

b) On a related note, using meta-analytic residuals to test for publication bias assumes that none of the modeled variables correlate with publication bias. Is that reasonable in this case?

We thank the reviewers for spotting a lack of clarity in our writing. To account for this comment, we added text (see response 2a above). The meta-analytic residuals used were those from the meta-analyses, which were intercept-only models where no other variables were modelled except the random effects. The models that include moderators are named “meta-regressions”, we used that nomenclature throughout.

c) If publication bias towards strong effects were at work, we would expect to observe the strongest effects from those studies with greater sampling error (those with lower sampling effort). The observed absence of this result is what we would expect with little or weak publication bias. This should be acknowledged (though see points a) and b) above).

We would indeed expect to observe the strongest effects when precision is low, which would lead to the funnel shape observed in Figure 2 and Appendix 2—figure 2. What we would expect in case of strong publication bias is asymmetry in the funnel plots, which our analyses do not seem to support. However, as noted above, detecting publication bias by visual inspection of funnel plots and running Egger’s regressions is difficult when the number of effect sizes is limited and heterogeneity is present (Moreno et al., 2009). Since heterogeneity is typically high in ecological and evolutionary meta-analyses (Senior et al., 2016), it is challenging to conclude whether publication bias may have existed. In our study, however, we were able to circumvent that problem and detect the existence of publication bias by using an alternative approach, i.e. by comparing published vs. unpublished effect sizes, and testing for the existence of time-lag bias.

We have clarified the most likely reason why we did not detect publication bias using funnel plots inspection and Egger’s regression tests by expanding the explanation we already had in the Discussion:

“Egger’s regressions failed to detect any funnel plot asymmetry, even in the meta-analyses based on published effect sizes only (Appendix 2—able 6). However, because unpublished data indeed existed (i.e. those obtained for this study), the detection failure was likely the consequence of the limited number of effect sizes available (i.e. low power) and the moderate level of heterogeneity found in this study (Sterne and Egger 2005; Moreno et al., 2009).”

In contrast, there is insufficient explanation of why the results in Figure 4 (the change in published effects over time) are strong evidence of publication bias.

We thank the reviewers for spotting a lack of clarity in our writing. We have now briefly explained in the text some of the processes that can lead to time-lag bias and referred the interested reader to an excellent review on the topic. Specifically, we have added the two following sentences to the manuscript.

“A decrease in effect size over time can have multiple causes. For example, initial effect sizes might be inflated due to low statistical power (“winner’s curse”) but published more easily and/or earlier due to positive selection of statistically significant results (reviewed by Koricheva, Jennions, and Lau, 2013).”

“An additional type of publication bias is time-lag bias, where early studies report larger effect sizes than later studies (Trikalinos and Ioannidis, 2005).”

3) Please state also the effect of using meta-analysis of the raw data, reanalysed in a consistent way, compared to using calculated effect sizes or summary statistics available in the published studies. While use of raw data does seem like a desirable standard to strive for in meta-analysis, it doesn't seem to make that big of a difference when comparing the results of 'published 1' and 'published 2' in Appendix—Table 6. Please comment whether you see this as a priority in a list of recommendations for best practice in meta-analysis (or best practice in the production of primary studies that can be effectively included in meta-analyses) or do the many other potential sources of bias have stronger effects on the confidence in/conclusions that can be drawn from meta-analytic estimates?

We thank the reviewers for these comments and suggestions. Theoretically, one of the most appealing features of a meta-analysis based on primary data is that, by analysing all the data in a consistent manner, effect sizes of all the studies are comparable (reviewed by Mengersen et al., 2013). From our analyses it is, however, difficult to conclude about whether meta-analyses based on primary data should be the preferred option. This is because our analyses were not designed to specifically test for differences between the two approaches. The main impediment for that was that we did not have access to the primary data of around half of the published studies (data available for 7 out of 13 studies), and therefore, there is still a substantial overlap between the meta-analyses “published 1” and “published 2” that might partly explain why the results of both meta-analyses did not differ much (Appendix 2—table6). Nevertheless, “published 2” estimated heterogeneity more precisely (i.e. narrower 95% CrI) than “published 1.

Lastly, we have no reason to believe that standardizing all effect sizes by re-analysing the primary data should lead to bias in the conclusions, but rather the opposite (see the response to the comment 2a above; see also a recent review about open data meta-analysis: Culina et al., 2018).

We reference two reviews about the topic in our Introduction (Simmonds et al., 2005; Mengersen et al., 2013) and we have now added a reference for a recent call to increase the use of meta-analysis of open datasets (Culina et al., 2018) (Introduction, third paragraph). Those three references provide strong support for the assertion that meta-analysis of primary data should be considered the gold standard.

4) It is necessary to add a statement (and explanatory text – compare https://osf.io/hadz3/) to the paper confirming whether, for all questions, you have reported all measures, conditions, data exclusions, and how you determined sample sizes.

We thank the reviewers for spotting this lack of transparency. We have now included the following sentence at the end of the Materials and methods section:

“We report all data exclusion criteria applied and the results of all analyses conducted in our study.”

See also our response to comment 1.

5) Please also address the following editorial point:The Abstract contains a statement about "the validity of the current scientific publishing culture". Similar statements are made in the main text (Introduction, last paragraph, Discussion, first and last paragraphs), but the manuscript never goes into detail about these matters.Please, therefore, delete the following passage from the Abstract: "raise important concerns about the validity of the current scientific publishing culture". Please also delete the corresponding statements in the last paragraph of the Introduction and the first paragraph of the Discussion. It is fine to keep the statement in the last paragraph of the Discussion.

We thank the editors for these suggestions, which we have now implemented.